# Single cell transcriptomic analysis of the immune cell compartment in the human small intestine and in Celiac disease

Nader Atlasy [1,10], Anna Bujko [2,7,10], Espen S. Bækkevold[2], Peter Brazda [1,8], Eva Janssen-Megens[1,9], Knut E. A. Lundin [3,4,5], Jørgen Jahnsen[6], Frode L. Jahnsen[2] & Hendrik G. Stunnenberg [1,8] ✉

Celiac disease is an autoimmune disorder in which ingestion of dietary gluten triggers an immune reaction in the small intestine leading to destruction of the lining epithelium. Current treatment focusses on lifelong adherence to a gluten-free diet. Gluten-specific CD4+ T cells and cytotoxic intraepithelial CD8+ T cells have been proposed to be central in disease pathogenesis. Here we use unbiased single-cell RNA-sequencing and explore the heterogeneity of CD45+ immune cells in the human small intestine. We show altered myeloid cell transcriptomes present in active celiac lesions. CD4+ and CD8+ T cells transcriptomes show extensive changes and we define a natural intraepithelial lymphocyte population that is reduced in celiac disease. We show that the immune landscape in Celiac patients on a gluten-free diet is only partially restored compared to control samples. Altogether, we provide a single cell transcriptomic resource that can inform the immune landscape of the small intestine during Celiac disease.

Celiac disease (CeD) is an autoimmune systemic condition, which manifests itself as a reaction of the immune system primarily against self-epithelial cells of the small intestine causing serious damage to the epithelium[1-4]. CeD affects about 1% of the population[5] and the affected individuals suffer from nutrition malabsorption, digestive tract pains and complications, and are at increased risk of developing cancer (2-4 fold for non-Hodgkin's lymphoma and >30 fold for small intestinal adenocarcinoma)[6,7]. CeD is caused by an immune response to ingested dietary gluten proteins, which are prolamin proteins from cereal grains such as wheat, rye and barley[8-11] leading to chronic inflammation of the small intestine, increased cell infiltration of lamina propria (LP) and intraepithelial lymphocytes (IELs), crypt hyperplasia, and villous atrophy[12]. Mucosal healing in CeD patients is a slow process, which can

take years in adults. The only available treatment for CeD patients is avoiding the exposure to gluten implying that the affected person has to adhere to a life-long gluten-free diet[13].

The genetic background plays a role in the onset of the disease and the vast majority of CeD patients possess specific variants of 'Major Histocompatibility molecule class II' (HLA class II) genes, HLA-DQ2 and HLA-DQ8[14]. However, these variants are not CeD specific; they are also present in 30–40% of the general population[15,16], only 2-3% of them develop the disease[1]. Previous studies show that upon gluten ingestion, the glutamine- and proline-rich large gliadin peptides resulted from the incomplete digestion of gluten molecules, enter the LP through epithelial transcellular or paracellular routes within the mucosal layer of the small intestine[17,18]. Furthermore, the enzyme

[1]Department of Molecular Biology, Faculty of Science, Radboud University, Nijmegen, The Netherlands. [2]Department of Pathology, University of Oslo and Oslo University Hospital, Rikshospitalet, Oslo, Norway. [3]KG Jebsen Coeliac Disease Research Centre, University of Oslo, Oslo 0372, Norway. [4]Institute of Clinical Medicine, University of Oslo, Oslo 0450, Norway. [5]Department of Gastroenterology, Oslo University Hospital Rikshospitalet, Oslo 0372, Norway. [6]Department of Gastroenterology, Akershus University Hospital and University of Oslo, Oslo, Norway. [7]Present address: VIB Center for Inflammation Research, B-9052 Ghent, Belgium; Department of Biomedical Molecular Biology, Ghent University, 9052 Ghent, Belgium. [8]Present address: Princess Maxima Centre for Pediatric Oncology, Heidelberglaan 25, 3584 CS Utrecht, The Netherlands. [9]Present address: NimaGen B.V., 6500 AB Nijmegen, The Netherlands. [10]These authors contributed equally: Nader Atlasy, Anna Bujko. ✉e-mail: H.G.Stunnenberg@prinsesmaximacentrum.nl

tissue transglutaminase (TG2), deamidates gliadin that is processed and subsequently presented by antigen-presenting cells (APCs) via HLA-DQ2 and HLA-DQ8 molecules to CD4[+] T cells[19]. This will result in a T cell immune response against the epithelial cells of the small intestine and at the same time auto-antibodies against gliadin, TG2 and actin are generated by B-cells which altogether will result in the intestinal and extraintestinal symptoms of the disease[20].

Although many efforts have been done to study the HLA-DQ-dependent CeD-associated T-cells and their clonal and functional properties[21–26], the detailed mechanistic pathways of the disease are not fully clear and moreover, most of the findings especially in human are based on analysis of heterogeneous populations of cells. Other approaches rely on pre-selection of canonical surface markers, which may mask the information about the heterogeneity of the cells within a population. A comprehensive-high resolution assessment of the immune cells in CeD versus healthy small intestine tissues is lacking. Here, we used single-cell RNA-sequencing combined with statistical modeling to comprehensively dissect the cellular heterogeneity of the

CD45[+] immune cells in the small intestine of CeD patients, patients adhering to gluten-free diet and healthy subjects in an unbiased manner.

## Results

### Single-cell RNA-seq analysis reveals five distinct immune-cell compartments in the LP of human small intestine

To assess the changes in the immune cell landscape of the human small intestine (hSI) in CeD we separated the epithelium from LP (see methods) and isolated CD45[+] immune cells separately from LP and the epithelial layer of duodenal biopsies obtained from newly diagnosed CeD patients ($n = 8$), patients who are on gluten-free diet (GFD) for at least one year ($n = 5$) and from control individuals without CeD (Ctrl) ($n = 7$). GFD and Ctrl patients showed normal histology (Marsh 0) (Fig. 1a and Supplementary Table 1). In line with previous studies[27,28], flowcytometry analysis of the markers used for index sorting shows that CeD harbors increased numbers of CD45[+] cells in LP and a slightly higher fraction of CD3[+] cells as compared to controls (Supplementary

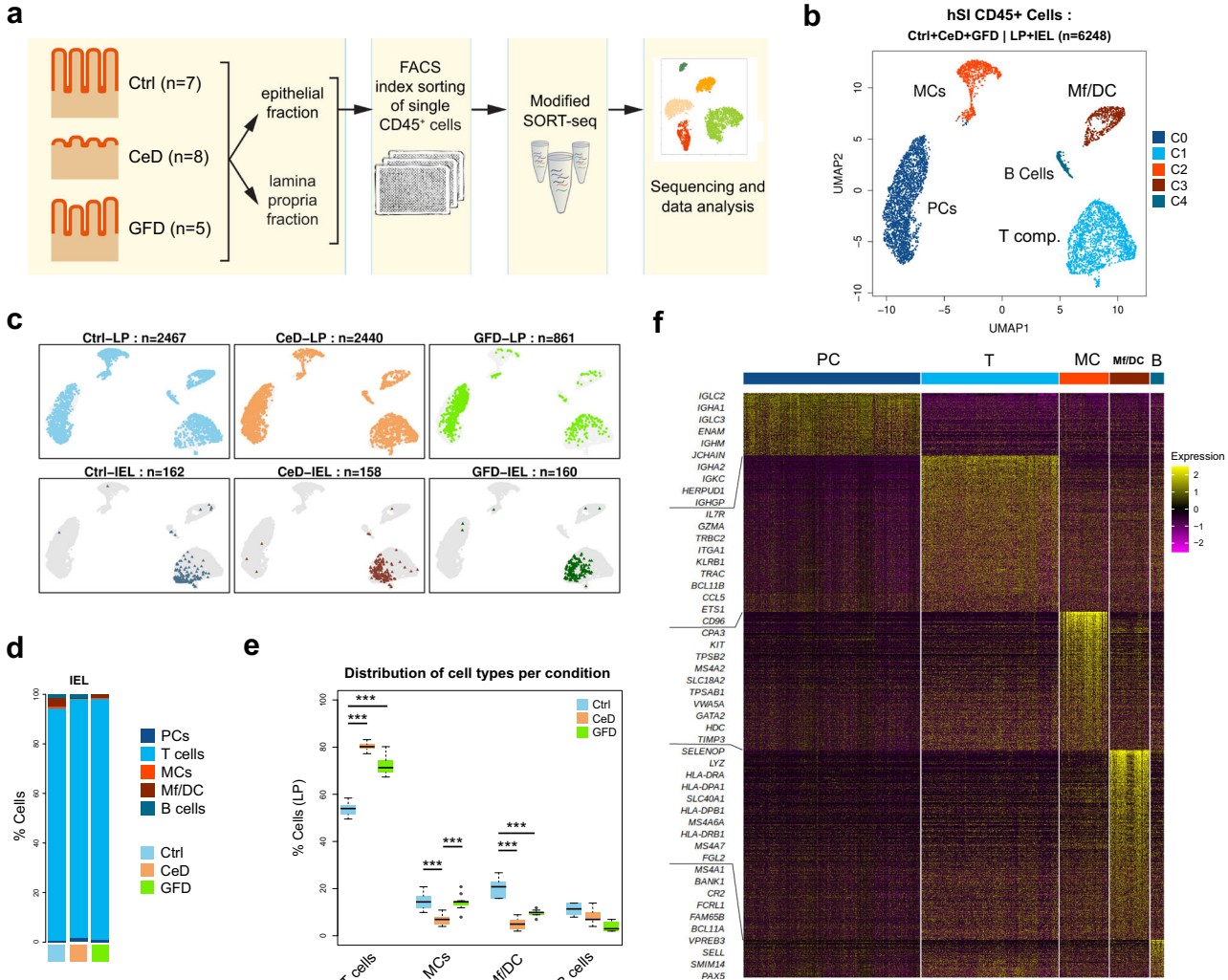

**Fig. 1 | Single-cell CD45[+] immune cells in human small intestine in Ctrl, CeD and GFD samples. a** Schematic study design **b** UMAP clustering scatter plot of hSI CD45[+] cells based on single cell mRNA expressions (PC Plasma cell, MC Mast cell, Mf/DC Macrophage/Dendritic cell; Ctrl control subjects, CeD Celiac disease subjects, GFD Celiac patients on gluten-free diet). **c** Projection of the donor status and the origin of the cells (LP vs IEL) on UMAP scatter plots **d** Bar plots depicting the proportion of the cell types detected within the epithelium of hSI per each condition in our single-cell RNAseq dataset **e** Boxplots showing the cell type related

percentage of the single cells among the CD45[+] cells for Ctrl, CeD and GFD derived cells (center line, median; box limits, upper and lower quartiles; whiskers, minimum and maximum values; points, outliers, 1.5x interquartile range; two-sided t-test $p$ value < = 0.05 (T: Ctrl vs CeD = $9.54 \times 10^{-9}$, Ctrl vs GFD = 0.0019; MCs: Ctrl vs CeD = $8.49 \times 10^{-5}$, CeD vs GFD = $1.90 \times 10^{-8}$; Mf/DC: Ctrl vs CeD = $2.73 \times 10^{-6}$, Ctrl vs GFD = $2.37 \times 10^{-5}$); $n = 10$ randomized cell subsampling). **f** Heatmap of differentially expressed genes of each cluster (cell type) showing the top-ranked genes.

Fig. 1a, b). Using a modified version of the SORT-seq[29] single-cell RNA-seq protocol and applying quality control cut offs yielded 6,248 single cells with high-quality RNA profiles (Fig. 1a and Supplementary Fig. 1c, d). *Uniform Manifold Approximation and Projection (UMAP)* analysis revealed five major immune-cell compartments that, based on classical lineage marker genes, were annotated as T-cells (CD3), plasma cells (PCs) (CD19 and *SDC1* (CD138)), macrophage/dendritic cells (MF-DC (CD14, CD11c and HLA-DR)), mast cells (MC) (*KIT*) and B cells (*MS4A1* (CD20)) (Fig. 1b and Supplementary Fig. 1e, f). Projecting the three conditions Ctrl, CeD and GFD on the UMAP clusters, we found that all major cell types were present in LP in each condition. Importantly in all three conditions the IELs are for the most part (>90%) localized to the T-cell compartment (Fig. 1c, d). Analysis of the proportion of the isolated immune cells in the three conditions (Ctrl, CeD and GFD) in LP revealed that in both CeD and GFD samples, the T-cell compartment was significantly enriched whereas MF-DCs were significantly less abundant as compared to Ctrl subjects. Moreover, we noticed a significantly reduced relative number of MCs in CeD when compared to Ctrl and GFD samples, ~5% in CeD compared to >15% in Ctrl and GFD (Fig. 1e).

We applied Wilcoxon rank sum test (as part of the 'Seurat' pipeline[30]) to identify cell type-specific transcripts distinguishing each cell population (Fig. 1f and Supplementary Data 1). We identified several genes that are differentially expressed in the CD45+ PC cluster with immunoglobulin-related genes such as *IGLC2, IGLC3, IGHA1* and *ENAM* amongst the top genes. For the T-cell compartment, genes including *IL7R, TRBC2, ITGA1* and *GZMA* and MC analysis yielded *CPA3, TPSB2, KIT* and *MS4A2* as the top-ranked genes. We found several marker genes distinguishing MF-DC cells with *SELENOP, LYZ, SLC40A1, MS4A6A* and *HLA* genes among the top ranked. B cells were differentially expressing markers with *MS4A1, BANK1* and *CR2* as top-ranked genes (Fig. 1f).

### Increased number of proinflammatory macrophages in CeD associated with INFg signaling

We have previously reported that healthy hSI harbors major populations of mature MFs that were hyporesponsive to inflammatory stimuli[31] as well as small subpopulations of immature proinflammatory MFs that are distinct from DCs. The current analysis of MF-DC cells from Ctrl, CeD and GFD revealed one major population of CD163 expressing MFs, with contributions from several donors (Supplementary Fig. 2a), and in addition three other small clusters (Fig. 2a). Importantly, we detected a small cluster of CD11c+ cells that was nearly exclusively found in CeD and that we classified as inf-MF cells based on the expression of inflammatory genes such as *VCAN, GBP1, LYZ, S100A4* and *WARS* (Fig. 2a, b-arrow and c). Based on the repertoire of signature genes, we identified the two other small clusters as cDC1 (expressing *XCR1* and *CADM1*) and as cDC2 (expressing CD1C and CD1E) (Fig. 2a, c). MFs were the predominant MF-DC population in all conditions with ~80% in both Ctrl and GFD samples and were significantly less abundant in CeD (~60%). The Inf-MF cells were exclusively more abundant in CeD samples (>20% of the MF-DC population) as compared to Ctrl and GFD (Fig. 2b). The DC populations were relatively equally present in all conditions with a slight reduction of DCs in CeD samples (Fig. 2b and Supplementary Fig. 2b). Differential expression analysis revealed that the cells of the major MF cluster express higher levels of mature signature genes including *SELENOP, MAF, SLC40A1, LGMN, NRP1, C1QB, ABCA1,* and *C1QC* compared to other MF-DC cells (Fig. 2c and Supplementary Data 1). cDC2 cells showed higher levels of *CD207, CD1C, FCER1A, CD1E* and *PLAC8* (Fig. 2c). cDC1 population consisted of a small number of cells (n = 19) that differentially express *CPNE3, CADM1, IDO1, XCR1* and *ENPP1* (Fig. 2c and Supplementary Data 1). UMAP analysis showed that the major MF cluster from CeD displays a gene distribution different from their counterparts in Ctrl subjects with many of the cells showing a *ITGAX* (CD11c)-high and *SELENOP/NRP1* and *C1QC*-low distribution

(Fig. 2d and Supplementary Fig. 2b). Additionally, flowcytometry analysis confirmed the over representation of CD11c+ over CD11c- cells in CeD samples as compared to Ctrl subjects (Supplementary Fig. 2b). These results are in line with earlier studies showing increased numbers of immature CD11+ monocyte-like MF in CeD lesions[32,33]. In Inf-MF cells, gene ontology analysis revealed a significant enrichment of biological processes involved in cytokine signaling, IFNg response and leukocyte activation with several member genes including *GBP1, GBP2, GBP4, GBP5, CD44* and *STAT1* that are highly expressed in these cells (Fig. 2c, e and Supplementary Data 1).

### MCs in hSI show distinct transcriptional programs in Ctrl, CeD and GFD

Within the MC compartment, UMAP analysis revealed four clusters of cells (C0-C3) (Supplementary Fig. 3a). CeD-MCs were mainly found in cluster C0, whereas GFD-MCs were more abundant in cluster C2 (Supplementary Fig. 3b) with no major donor batch effect (Supplementary Fig. 3c). We found that cells in CeD-dominated cluster C0 had higher expression of genes including *TPSAB1, CADPS* and *PLAT*. Cluster C1 cells differentially express genes like *EGR1, CTSG, FOS* and *NFKBIA*. Cells in GFD-dominated C2 showed higher expressions of *PAN3, DNMT3A* and *JCHAIN* among others. The smallest cluster C3 differentially expressed top genes including *ATP8A1, FBXO22* and *SERPINB13* (Supplementary Fig. 3d and Supplementary Data 1).

Gene ontology analysis on cluster markers revealed that CeD-dominated C0 cells are associated with biological processes such as 'protein to ER process', 'antigen processing and presentation' and 'positive regulation of T cell-mediated cytotoxicity'. Cluster C1 that is predominantly formed by Ctrl cells is associated with biological processes related to 'myeloid cell activation' and 'neutrophil activation'. In GFD-dominated cluster C2, the cells were associated with 'humoral immune response' and 'positive regulation of B cell activation' biological processes (Supplementary Data 1).

### T-cell compartment changes with disease state

UMAP clustering analysis of the T-cell compartment in LP (Fig. 1a) revealed the two main clusters of CD4+ and CD8+ T-cells (Fig. 3a). In addition, we found smaller clusters of natural killer cells (NKs) expressing *GNLY* and *NKG7* as well as *KIT*-expressing innate lymphoid cells (ILCs). A small group of proliferating cells was also detected (Fig. 3a). Comparison of the distribution of the cell types between conditions revealed that CD4+ T cells were the most abundant cells in control donors (>70%) whereas in CeD and also in GFD, we noticed a higher presence of CD8+ T cells comparable to CD4+ T cell number (~50%) (Fig. 3b). Projection of the FACS protein intensity analysis on the UMAP clusters showed that the NKs and ILCs were very low in CD3 expression as compared to T cells (Fig. 3c, arrows). Within the distinct NK cluster, a small number of cells expressed CD3 at a level comparable with T cells thus resembling NKT cells[34] (Fig. 3c). Using Wilcoxon rank sum test analysis, we identified differentially expressed genes between the cell types (Fig. 3d). CD4+ T cells highly expressed *IL7R* and *MAF* whereas CD8+ T cells differentially expressed *CCL5, ITGAE* and *CD160*. The CD8+ T cell cluster also contained a large fraction of cells expressing *TRGC2* and *TRGC1* associated with g/d T cells. Cells in the ILC[35] cluster differentially expressed genes such as *PCDH9, KIT, ALDOC* and *AHR*. Expression analysis of the GWAS-associated genes containing CeD-related SNPs obtained from a publicly available catalog (The NHGRI-EBI Catalog of human genome-wide association studies-EFO_0001060) showed complex patterns occurring in both CeD and GFD cells (Fig. 3e). In CD4 T cells, *CCR2* and *PUS10* are lower and *IL2RA* is relatively higher expressed in both CeD and GFD cells when compared to Ctrl. In CD8 T cells, *PRKCQ, ADGRL2* and *HLA-DQA1* genes are lower and *IRF4* and *RUNX3* are higher expressed in both CeD and GFD as compared to Ctrl cells. In NK cells, genes such as *MTTP, NKD1, YDJC* and *NFIA* are lower and

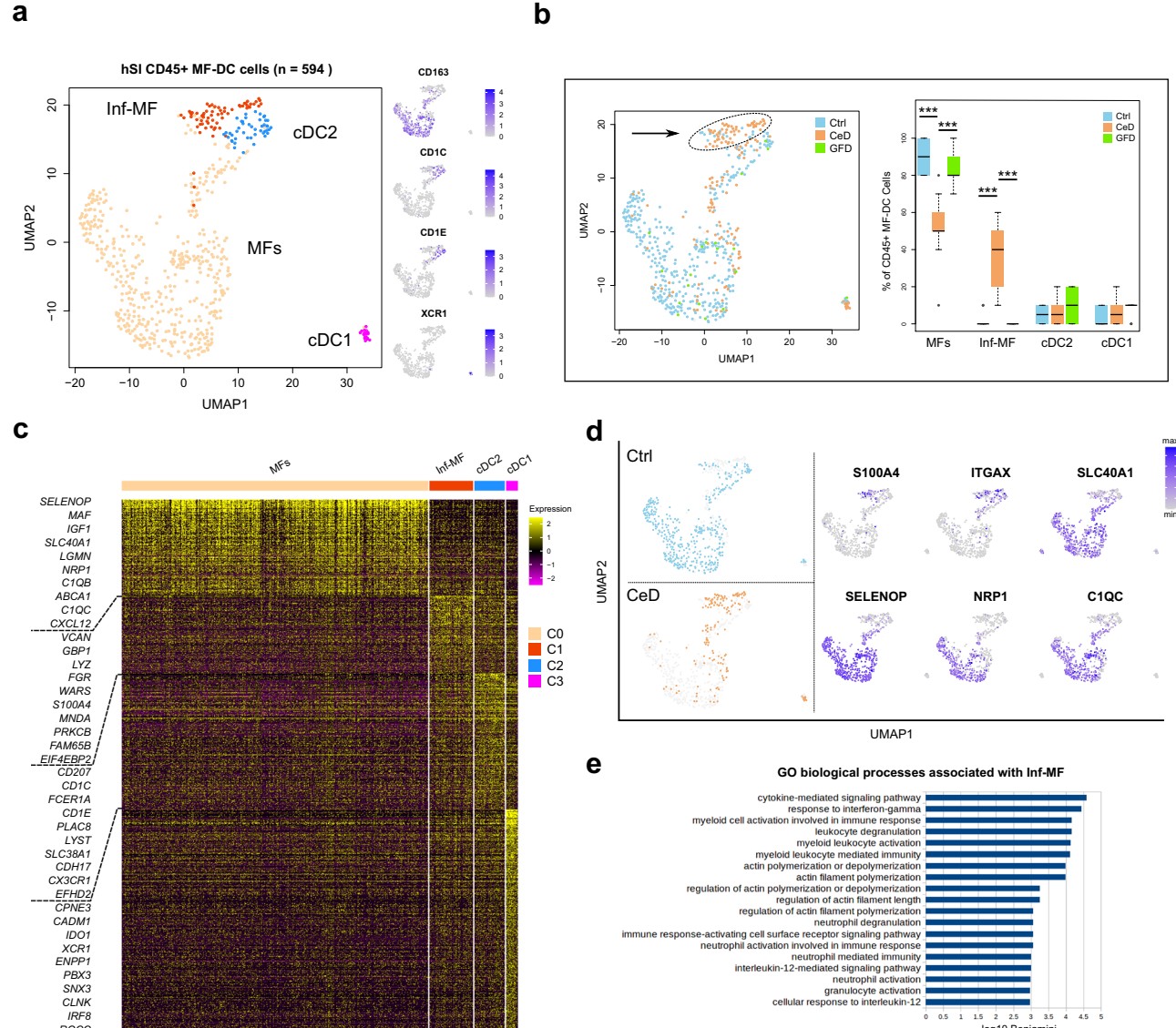

**Fig. 2 | Single-cell landscape of MF-DC cells in Lamina Propria of human small intestine in Ctrl, CeD and GFD samples. a** UMAP clustering plot of CD45⁺ Macrophage/Dendritic (MF-DC) cells in Lamina Propria (LP) of Ctrl (control subjects), CeD (Celiac disease patients) and GFD (Celiac patients on gluten-free diet) samples with side scatter plots of canonical gene expressions (MF Macrophage, Inf-MF Inflammatory Macrophage, cDC classical Dendritic cell). **b** Projection of Ctrl, CeD and GFD cells on the UMAP plot (left scatter plot) and boxplots (right) showing the percentage of the MF-DC single cells across the UMAP clusters from Ctrl, CeD and GFD derived cells (center line, median; box limits, upper and lower quartiles;

whiskers, minimum and maximum values; points, outliers, 1.5x interquartile range; two-sided t-test $p$ value < = 0.05 (MFs: Ctrl vs CeD = $3.54 \times 10^{-8}$, CeD vs GFD = $6.32 \times 10^{-7}$; Inf-MF: Ctrl vs CeD = $1.46 \times 10^{-7}$, CeD vs GFD = $9.15 \times 10^{-7}$); $n = 10$ randomized cell subsampling). **c** Heatmap of the differentially expressed genes between the myeloid UMAP clusters. **d** Scatter plots depicting distribution of the cells between Ctrl and CeD samples together with expression pattern of selected MF genes projected on the UMAP. **e** Barplots of top biological processes from gene ontology analysis on differentially expressed genes in Inf-MF cluster (Benjamini FDR < 0.009).

*MYNN* is higher expressed in CeD/GFD as compared to Ctrl. In ILC cells, several genes including *PPP1R12B*, *GLB1*, *SUOX* and *DAG1* are lower and *PTPN2* and *BACH2* are higher expressed in both CeD and GFD cells when compared to Ctrl counterparts (Fig. 3e). A significant contribution to the overall genetic predisposition to CeD is due to non-DQ HLA genes[36] and our data aid to disentangle the complex genetics of CeD and the impact on the gut immune system.

### *IKZF2 / CD39* expressing *CD4⁺* T cells are significantly increased in LP of CeD patients

Within the CD4⁺ T cell population in LP, we found two distinct clusters of cells (C0 and C1) (Fig. 4a). The majority of the CD4 T-cells (~90% in Ctrl and GFD and ~50% in CeD) were in cluster C0. In CeD patients, the number of C1 cells was significantly higher (>40%) compared to Ctrl

and GFD subjects (~10%) (Fig. 4b and Supplementary Fig. 4a). Top differentially expressed genes in cluster C0 included *ITGA1, MGAT4A, SORL1, TC2N* and *ERN1*. The CeD-dominated cluster C1 cells significantly expressed genes including *TBC1D4, IKZF2, TIGIT, LEF1, MAF, TFRC* and *ENTPD1 (*CD39*)* (Fig. 4c). This phenotype partly resembles that reported for gluten-responsive CD4⁺ T cells in the blood and intestines of CeD patients expressing activation markers including CD39 and regulatory T cell (Treg) markers such as *IKZF2* and *TIGIT*[37,38]. Analyzing mRNA expression of the constant region of TCR genes, we found that the vast majority of CD4⁺ T cells are TCRa/b expressing cells (Supplementary Fig. 4b) in agreement with previous studies on clonotype analysis of disease-driving CD4 T cells in CeD[25]. Gene ontology analysis of the CeD-dominated cluster C1 marker genes revealed 'cytokine production and signaling' related biological processes.

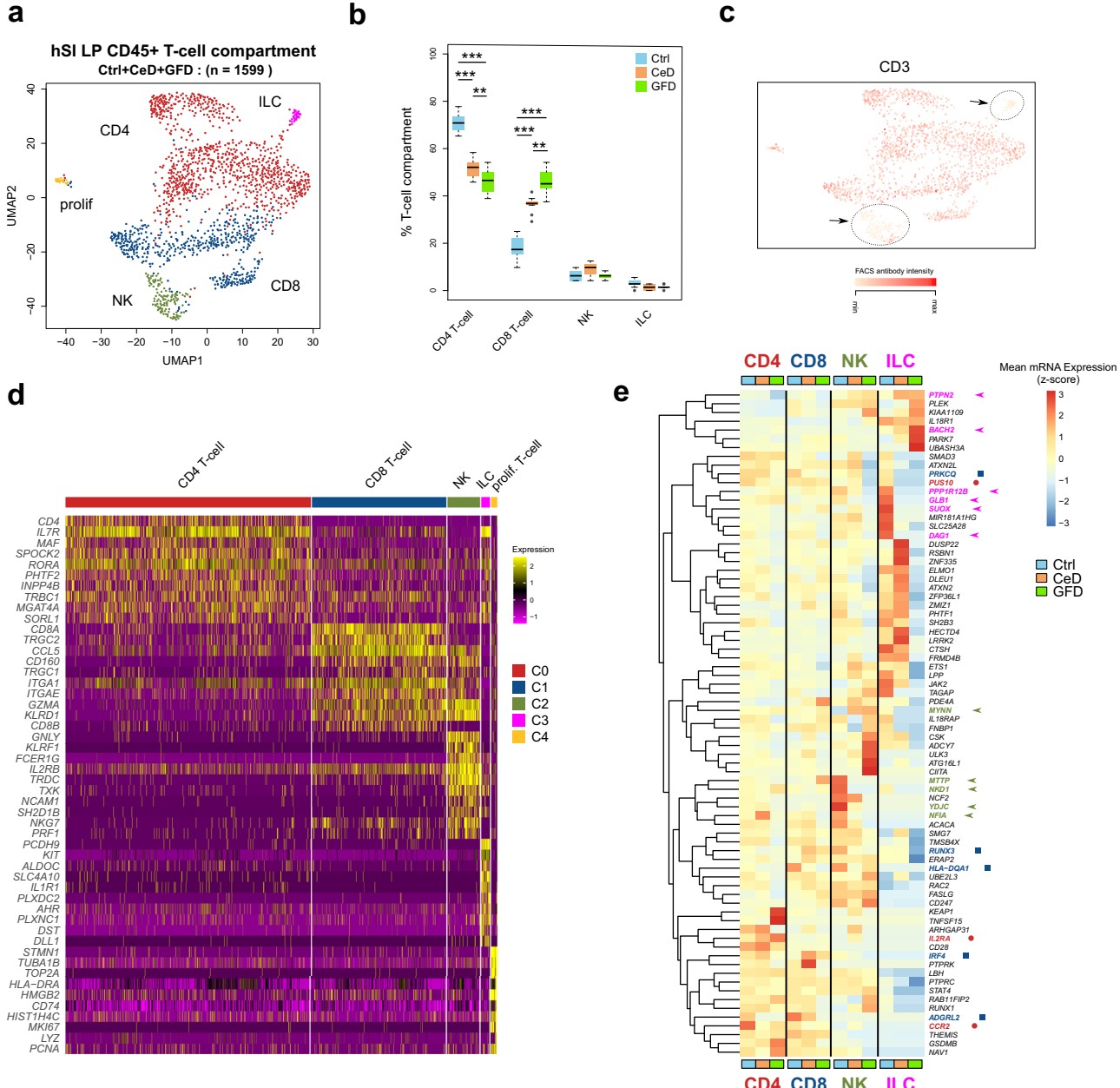

**Fig. 3 | Single-cell landscape of T-cell compartment in human small intestine (Lamina Propria) in Ctrl, CeD and GFD samples. a** UMAP clustering plot of single-cell RNA-seq transcriptome profiles of cells in T-cell compartment (as in Fig. 1b) depicting CD4+ and CD8+ T cells, Innate lymphoid cells (ILCs) and Natural killer cell (NK) clusters (prolif proliferating cells). **b** Boxplots depicting the percentage of the different cell types among the total number of the T-cell compartment across the UMAP clusters of Ctrl (control subjects), CeD (Celiac disease patients) and GFD (Celiac patients on gluten-free diet) samples (center line, median; box limits, upper and lower quartiles; whiskers, minimum and maximum values; points, outliers, 1.5x

interquartile range; two-sided t-test $p$ value < = 0.05 (T-CD4: Ctrl vs CeD = $2.90 \times 10^{-7}$, Ctrl vs GFD = $1.45 \times 10^{-12}$, CeD vs GFD = 0.0088; T-CD8: Ctrl vs CeD = $3.70 \times 10^{-9}$, Ctrl vs GFD = $1.07 \times 10^{-11}$, CeD vs GFD = 0.0019); $n = 10$ randomized cell sub-sampling). **c** Projection of the FACS intensity signal of the anti-CD3 antibody on the T-cell UMAP clusters showing the CD3-low clusters (arrows). **d** Heatmap of differentially expressed genes between the CD4+ T, CD8+ T, NK and ILC UMAP clusters showing the top-ranked genes. **e** Heatmap of the CeD-associated genes obtained from publicly available GWAS catalog showing the expression patterns between Ctrl, CeD and GFD across the T-cell compartment.

Moreover, DEG in C1 are associated with several signaling pathways including 'NOD-like receptor signaling pathway', 'TCR signaling' and 'NF-kappa B signaling pathway' among the top terms in KEGG pathway analysis (Fig. 4d). Among the top enriched KEGG pathways of cluster C1 cells, we found several infection-related pathways that were upregulated, which could be associated with an activated state in response to exposure to gluten molecules and/or viral infections or due to gut microbiome alterations in CeD[39] (Supplementary Fig. 4c).

We assessed the mRNA expressions of the cytokine/chemokine and their associated receptors in the CD4+ T clusters. Our analysis

revealed that the CeD-associated cluster C1 cells express higher levels of *IL32* and *TNFAIP3* and they upregulate *IL2RB* and *IL6ST* expression but displayed lower expression of *CCL5*, *TGFBR2* and *IL7R* genes as compared to cluster C0 cells (Fig. 4e, f).

**LP CD8+ T cells display an activated phenotype in CeD patients**
The CD8+ T cell population in the LP clustered into four clusters (C0 to C3) (Fig. 5a), that had contributions from several donors with no major observed donor batch effect (Supplementary Fig. 5a). We found that the majority of the CD8+ T cells in Ctrl are in cluster C1

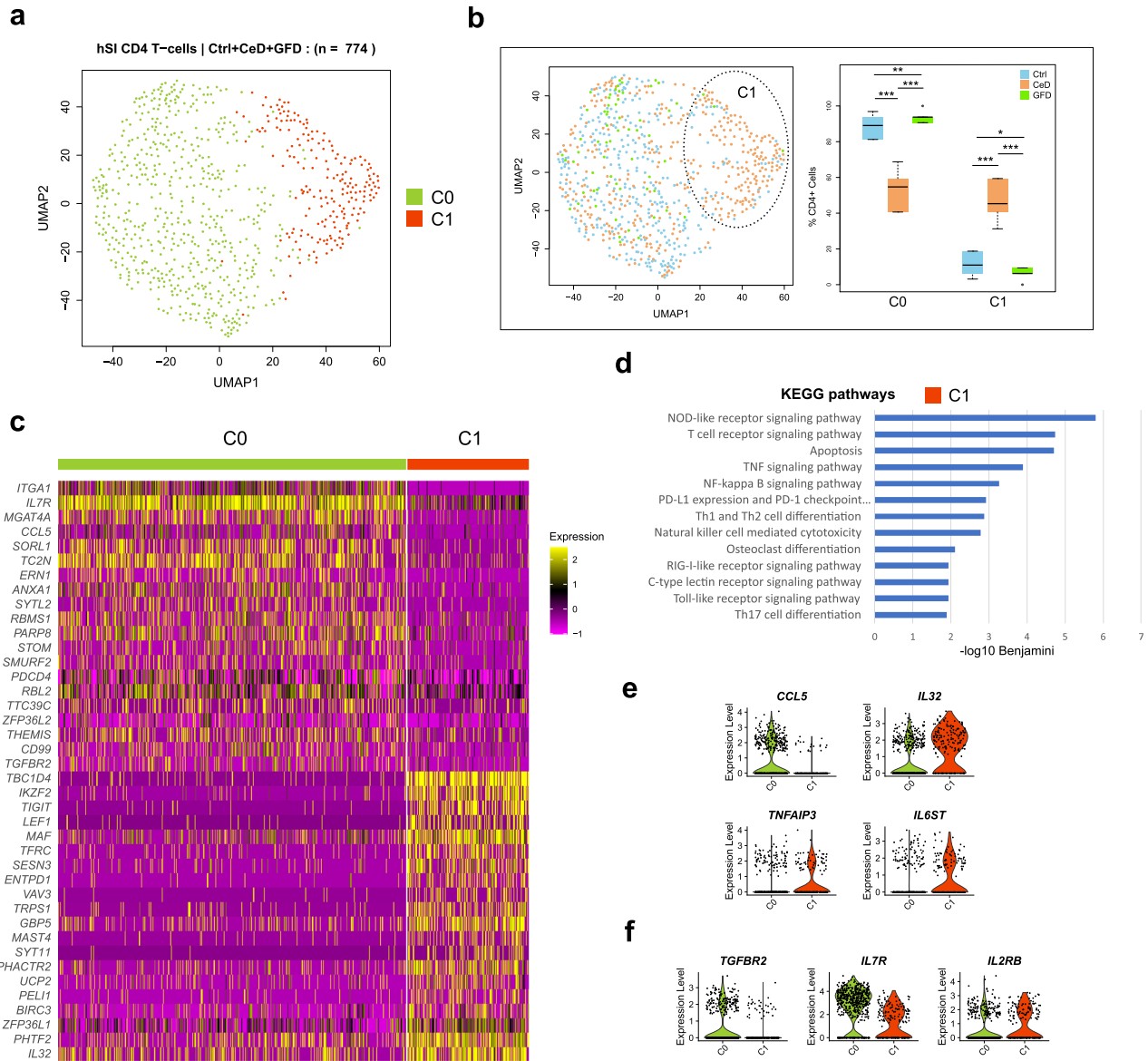

**Fig. 4 | Single-cell landscape of CD4⁺ T cells in Lamina Propria of human small intestine in Ctrl, CeD and GFD samples. a** UMAP clustering scatter plot of CD4⁺ T cells in Lamina Propria (LP) of Ctrl (control subjects), CeD (Celiac disease patients) and GFD (Celiac patients on gluten-free diet) samples **b** Projection of Ctrl, CeD and GFD derived cells on the UMAP clusters (left scatter plot) and the percentage of cells across the clusters per each condition as compared to each other (right boxplots) (center line, median; box limits, upper and lower quartiles; whiskers, minimum and maximum values; points, outliers, 1.5x interquartile range; two-sided t-test *p* value < = 0.05 (C0: Ctrl vs CeD = 4.38 × 10⁻⁹, Ctrl vs GFD = 0.0038, CeD

vs GFD = 2.25 × 10⁻⁸; C1: Ctrl vs CeD = 7.75 × 10⁻⁸, Ctrl vs GFD = 0.0269, CeD vs GFD = 1.26 × 10⁻⁷); *n* = 10 randomized cell subsampling) **c** Heatmap of the top differentially expressed genes showing the transcriptional difference between the CeD-dominated cluster C1 and the rest of CD4⁺ T-cells (C0) in the LP of hSI. **d** Bar plots resulted from gene ontology analysis of differentially up-regulated genes in CeD-dominated CD4⁺ T-cell cluster C1 as compared to other CD4⁺ T-cells (C0) showing the top associated KEGG pathways. **e** Violin plot showing the differentially expressed cytokine ligands and **f** the differentially expressed cytokine receptors among CD4⁺ T cells.

(~50%) whereas the cells in GFD samples were mainly distributed between C1 and C3. In CeD patients, the CD8⁺ T-cells strongly dominated cluster C0 (Fig. 5b). Differential expression analysis using Bayesian modeling revealed that cluster C0 cells highly express *ITGAE (CD103), ENTPD1 (CD39), ID2, RBPJ, GBP5* and *PRRG3* amongst others, indicating the activated state of these tissue-resident cells[37,40]. In gene ontology analysis, we found several infection-related pathways associated with the cells in cluster C0 (Supplementary Data 1). Cluster C1 cells differentially express *CD8A, B2M, BCL11B, KLRB1* and *TPT1* genes, in which CD8⁺ / *KLRB1*⁺ was shown to be associated with a cytotoxic memory phenotype[41]. Cells in cluster C2 differentially express *GZMK, AHNAK, KLRG1, PLEK* and *CD44* that in accordance with the previous findings suggests that they are recently recruited

CD8⁺ / KLRG1⁺ T cells[40,42] (Fig. 5c). Lastly, cells in cluster C3 differentially express *IKZF2, TRDC, KIR2DL4, ATP8B4 and PDE3B* as top genes. These cells also highly express *ITGA1 (CD49a)* that is a marker of tissue-resident memory (T_RM) cells[43–45]. We used the mRNA expression level of the respective TCR constant regions to assess the TCRg/d and TCRa/b expressing cells. Our results showed that TCRg/d expressing cells were mainly located in cluster 3 (Supplementary Fig. 5b).

Assessment of mRNA expression of cytokine/chemokine and their associated genes showed that the CeD-dominated cluster C0 cells differentially express higher levels of *IL16, IL18BP, CCL5, CCL20* and *IFNG* and receptors *IL18R1, IL10RB, CCR9, CXCR6, IFNAR2, TNFRSF1B, TNFRSF17* and *TNFRSF9*. Interestingly we found that *IL32*, which was

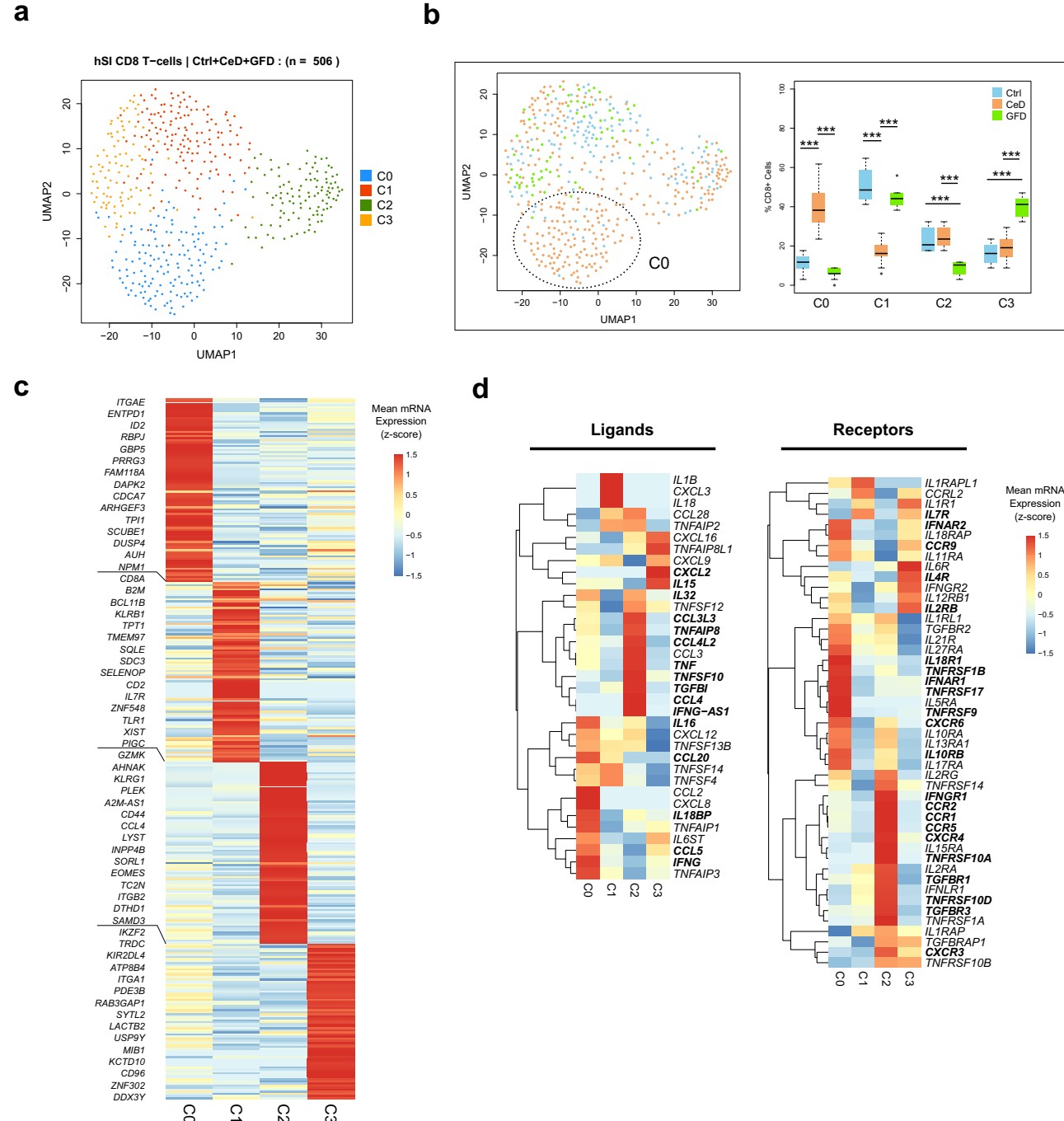

**Fig. 5 | Single-cell landscape of CD8⁺ T cells in Lamina Propria of human small intestine from Ctrl, CeD and GFD samples. a** UMAP clustering depicting the heterogeneity of CD8⁺ T cells in Lamina Propria (LP) from Ctrl (control subjects), CeD (Celiac disease patients) and GFD (Celiac patients on gluten-free diet) samples. **b** Projection of cells according to Ctrl, CeD and GFD origin on the UMAP clusters (left scatter plot) and the boxplots showing the distribution of cells across the clusters per each condition as compared to each other (right) (center line, median; box limits, upper and lower quartiles; whiskers, minimum and maximum values; points, outliers, 1.5x interquartile range; two-sided t-test $p$ value < = 0.05 (C0: Ctrl vs

CeD = $3.45 \times 10^{-8}$, CeD vs GFD = $1.30 \times 10^{-8}$; C1: Ctrl vs CeD = $7.15 \times 10^{-7}$, CeD vs GFD = $3.14 \times 10^{-9}$; C2: Ctrl vs GFD = $4.56 \times 10^{-5}$, CeD vs GFD = 0.0004; C3: Ctrl vs GFD = $1.51 \times 10^{-7}$, CeD vs GFD = $7.38 \times 10^{-8}$); $n = 10$ randomized cell subsampling). **c** Heatmap plot of the mean expression of top differentially expressed genes per each CD8⁺ T-cell cluster (empirical Bayes two-sided $p$ value < = 0.05). **d** Heatmaps depicting the gene expression of the cytokine ligands (left) and receptors (right) between the clusters of CD8⁺ T cells (differentially expressed genes per each cluster are shown in bold).

highly expressed in CeD-associated *CD4*⁺ T cells, is also upregulated in cluster C0 cells suggesting a potential role in the course of the disease. Cells in cluster C2 highly express *CCL3L3, CCL4L2, CCL4, TNFAIP8, TNF, TNFSF10* and *TGFBI* and the receptor genes *CCR1, CCR2, CCR5, CXCR3, CXCR4, IFNGR1, TNFRSF10A, TNFRSF10D, TGFBR1* and *TGFBR3*. T_RM cells in cluster C3 had higher expressions of *IL15* and *CXCL2* and

receptor genes *IL4R* and *IL2RB* (Fig. 5d, left heatmap (ligands); right heatmap (receptors)). Given the relatively higher expression of *IL15RA* in cluster C2 cells, the T_RM cells (C3) in CeD patients on gluten-free diet may play a role in stimulating the peripheral blood lymphocytes in LP and hence enhancing the CD8⁺ T cell response upon gluten stimulation.

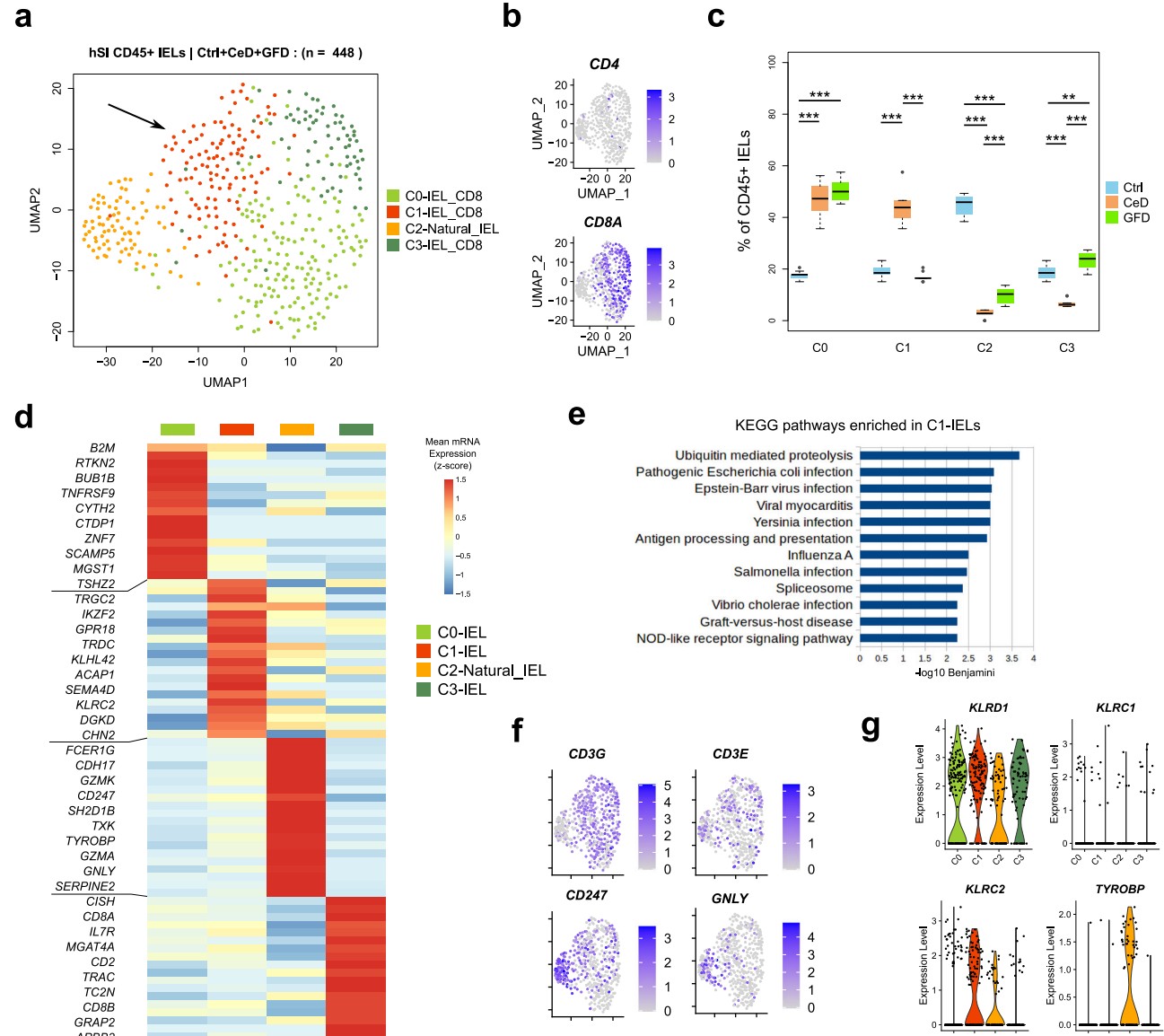

**Fig. 6 | Single-cell landscape of CD45⁺ Intraepithelial lymphocytes of human small intestine in Ctrl, CeD and GFD samples. a** UMAP clustering plot of hSI CD45⁺ Intraepithelial lymphocytes (IELs) derived from the epithelium of the Ctrl (control subjects), CeD (Celiac disease patients) and GFD (Celiac patients on gluten-free diet) samples. **b** Scatter plots depicting the *CD4* and *CD8A* mRNA expression across the IELs from Ctrl, CeD and GFD samples on the UMAP plot. **c** Boxplots showing the distribution of IELs across the UMAP clusters per each condition (Ctrl, CeD and GFD) as compared to each other (center line, median; box limits, upper and lower quartiles; whiskers, minimum and maximum values; points, outliers, 1.5x interquartile range; two-sided t-test *p* value < = 0.05 (C0: Ctrl vs CeD = 2.27 × 10⁻¹²,

Ctrl vs GFD = 5.43 × 10⁻¹³; C1: Ctrl vs CeD = 1.36 × 10⁻¹², CeD vs GFD = 3.84 × 10⁻¹²; C2: Ctrl vs CeD = 2.24 × 10⁻¹⁰, Ctrl vs GFD = 6.34 × 10⁻¹¹, CeD vs GFD = 1.08 × 10⁻⁶; C3: Ctrl vs CeD = 2.19 × 10⁻⁹, Ctrl vs GFD = 0.0032, CeD vs GFD = 2.12 × 10⁻⁹); *n* = 10 randomized cell subsampling). **d** Heatmap of the mean expression per cluster of top differentially expressed genes between the IEL clusters (empirical Bayes two-sided *p* value < = 0.05). **e** Bar plots showing the top enriched KEGG pathways in CeD-dominated cluster C1 cells in gene ontology analysis. **f** Scatter plots depicting the mRNA expression of selected genes on the UMAP plot in IEL populations. **g** Violin plots showing the expression of DAP12 (*TYROBP*) signaling selected genes in IELs.

## Elevated numbers of activated intraepithelial T cells and a reduced population of natural IELs in CeD patients

As part of the study design, we also directly isolated and analyzed specifically the CD45⁺ IELs from the same donors (In total 448 high quality single cells). UMAP analysis resulted in four clusters of cells with the clusters C0, C1 and C3 consisting of CD8⁺ T cells. The clusters were formed by several donors with no batch effect (Supplementary Fig. 6a). *CD8A* or *CD4* mRNA expressions was not detectable in cluster C2 cells (Fig. 6a, b). We found that both in CeD and in GFD donors, around 50% of the IELs were in cluster C0 and these cells do not disappear in GFD patients even with adhering to a gluten-free diet. Cluster C1 was predominantly formed by CeD cells (Fig. 6c). Cluster C2 largely consisted of cells from Ctrl subjects (~50%), in GFD patients the

proportion of these cells remain relatively low. Cluster C3 cells were reduced in CeD but restored in GFD patients (Fig. 6c). We did not find a significant pattern that separates the TCRa/b and TCRg/d populations in relation to the identified clusters of IELs. However, we do observed a higher presence of TCR g/d expressing cells within the CeD-dominated C1 which is in line with the previous findings that the TCR g/d IELs are increased in CeD⁴⁶ (Supplementary Fig. 6b).

Differential expression analysis between the clusters using Bayesian modeling revealed that cells in cluster C0 significantly express *BTM, RTKN2, BUB1B, TNFRSF9* and *CYTH2* as top genes (Fig. 6d). The CeD-dominated cluster C1 differentially expressed genes including *TRGC2, IKZF2, GPR18, TRDC* and *KLHL42* that are associated with 'infection' related response pathways including 'NOD-like receptor

signaling pathway' in gene ontology analysis (Fig. 6e). The cells in cluster C1 also differentially express *KLRC2* (NKG2C), a NK lineage receptor that has been previously shown to be involved in CeD pathology[47]. Our analysis showed that even though the donors followed a gluten-free diet, the CeD-associated intraepithelial CD8⁺ T cells (Fig. 6a, cluster C1) were still present in the GFD subjects with variable proportions (Supplementary Fig. 6a). Cells in cluster C2 significantly expressed genes including *FCER1G, CDH17, GZMK, CD247* and *SH2D1B* of which many genes are associated with the natural IELs as previously reported in mice[48] (Fig. 6d). Cells in cluster C3 differentially express *CISH, CD8A, MGAT4A* and *CD2* amongst others indicating their effector phenotype (Fig. 6d). Cytokine gene expression analysis revealed that CeD-dominated cluster C1 cells highly express *IL32, IFNG* and *CCL3L3* and the receptor genes *IL4R, IL10RA, IL10RB, TGFBRAP1, TGFBR2, TGFBR3, IFNAR2* and *TNFRSF1A*. Natural IELs in cluster C2 express high levels of *IL15, IL18, CCL28, CXCL2, CXCL8, CXCL14, CXCL16, TGFBI, TNFAIP2* and *TNFSF15* and the receptor genes *IL18R1, IL18RAP* and *CCR1*. Finally, cells in cluster C3 highly express *IL16, CCL5, CXCL9, TNF* and *TNFAIP8* and the receptor genes *IL2RG, IL7R, IL15RA, IL21R, IL27RA, CCR5, CCR9, CXCR4, TNFRSF10A* and *TNFRSF13C* (Supplementary Fig. 6c).

Natural IELs in our dataset had low expressions of *CD3* and high expression of *GNLY* genes (Fig. 6f), a phenotype that is associated with NK cells. To assess their phenotype, we directly compared the natural IELs with all other CD4⁺ and CD8⁺ T cells, NKs and ILCs from LP along all the IELs in our dataset. UMAP clustering analysis showed that the natural IEL formed their own cluster (Supplementary Fig. 6d) and that they possess a unique signature different from LP NK, ILC and T cells (Supplementary Fig. 6e). Assessment of the DAP12 receptor signaling axis, a NK cell characteristic associated with IE-CTLs in CeD[47], revealed that the CeD-associated cells in cluster C1 display high levels of *CD94 (KLRD1)/NKG2C (KLRC2)* receptors and are devoid of *NKG2A (KLRC1)* (Fig. 6g). DAP12 (*TYROBP*) gene was, however, exclusively expressed in natural IELs but not in the cluster C1 IELs, suggesting an alternative signaling in CeD (Fig. 6g). Interestingly, we found that natural IELs in cluster C2 express high levels of *CD94/NKG2C* genes but to a lower extent as compared to C1.

## Comparison of single-cell results with available studies on tissue bulk transcriptome in CeD

We compared our findings on immune landscape heterogeneity with two recent publicly available bulk RNA sequencing studies on CeD. We assessed the gene signature of differentially upregulated genes in tissue biopsies from GFD patients post-gluten challenge (PGC) by *Dotsenko* et al.[49], and the signature of highly upregulated genes in Celiac (PED = CeD) mucosa as compared to healthy controls (CTR = Ctrl) by *van der Graaf* et al.[50]. Our analysis showed that the upregulated genes in both tissue signatures are mostly associated with MF-DC and MC clusters with relatively less contribution from T cells and IELs in CeD patients in our dataset (Supplementary Fig. 7a, b). Using deconvolution analysis and our single-cell transcriptome signatures of the clusters, we then investigated the proportional state of the identified single-cell clusters in the *van der Graaf's* tissue dataset. This analysis revealed the CeD-related changes in the main clusters/cell types e.g. increased percentage of Inf-MFs and CD4⁺ T cluster C1, and decreased proportion of MFs and CD4⁺ T cluster C0 in CeD as compared to Ctrl. Beside subtle overall increase in CeD cells, the deconvolution analysis did not result in a relevant readout for the LP CD8⁺ T cells and the IELs (Supplementary Fig. 7c). We assessed our identified natural IEL signature with the *van der Graaf's* bulk RNAseq dataset[50] and found that several genes such as *CD247* and *CDH17* were upregulated in healthy tissue (CTR) while some genes i.e *ELP6* and *SRSF1* were upregulated in celiac mucosa (PED) (Supplementary Fig. 7d). The latter might indicate a shared gene expression with other stromal cells in the tissue.

## Discussion

CeD is a complex immune disorder with multifaceted mechanisms that not only affects the digestive tract locally but also manifests itself with systemic symptoms[51,52]. This involves an interplay between different immune cells from both innate and adaptive systems to which initiates and promotes the course of the disease. Hence, understanding and characterizing the main players of the immune system in CeD is of great interest and importance.

In this study, we provide a high-resolution comprehensive database and document prominent changes in the transcriptome landscape of the immune cells in the CeD lesions. Within innate immune cells, we show that mature MFs are greatly reduced in CeD in accordance with previous observations[33,53]. Several studies have shown a strong mucosal response to cytokines, including IFNg in the CeD[54–56]. We show that the Inf-MFs are more abundant in CeD patients and their transcriptome program is significantly associated with response to IFNg in LP. We hypothesize that the strong IFNg signaling in LP, might play a central role in the suppression of MF maturation[57] and accumulation of proinflammatory MFs in CeD[58,59]. Our transcriptome analysis revealed the prevalence of biological differences in MCs between Ctrl, CeD and GFD conditions suggesting that they may play a role in the course of the disease[60]. This could open more avenues to further investigate the involvement of MCs in CeD.

We found that the number of CD3⁺ T cells is increased in CeD and that the majority of CD4⁺ and CD8⁺ T cells are transcriptionally different from their control counterparts and from GFD. Previous studies on CeD were focused on gluten-specific T cells, which accounts for a minor subset[61]. However, in our study, we find that there is a dramatic shift in the transcriptome of a large population of both CD4⁺ and CD8⁺ T cells that are not necessarily gluten-specific, suggesting that a large population of both cell types are involved in the disease. We show that the CeD-associated CD4⁺ T cells are transcriptionally associated with several infection-related pathways, which could be a result of intestinal microbiome alterations in CeD patients[39,62]. These CD4⁺ T cells highly express *TIGIT* and *IKZF2* (*Helios*) indicating a Treg phenotype[63,64]. *Helios* has been associated with T$_H$2 differentiation[65] and it is differentially expressed in regulatory T (Treg) subpopulations resulting in different phenotypes and functions of the cells[66,67]. The CeD-associated CD4⁺ T cells express high levels of *CD39*, which is in accordance with previous studies showing their involvement in autoimmune diseases including CeD[61,68]. Our findings suggest that CD4⁺ T cells in active CeD, possess a transcriptionally mixed phenotype with activated cells and cells with regulatory properties. LP CD8⁺ T cells have received little attention as contributors to CeD pathology. Here we find that a large fraction of LP CD8⁺ T cells express an activated transcriptome program, suggesting that cells may play a more central role in CeD pathology than hitherto anticipated[69]. It has been shown that interferon-I signaling activates conventional dendritic cells to promote MHCI-mediated CD8⁺ T immunity[70]. This raises the notion that the CeD-associated LP CD8⁺ T cells might be activated prior to migration into the LP of hSI through MHCI cross-presentation[71] and/or they might have a dynamic turnover between the LP and the epithelium of hSI in CeD. Within the IELs, we identified a natural IEL population, previously reported in mice studies[48], in the epithelium of the hSI that display a distinct transcriptome signature distinguishing it from other cells within the T- cell compartment (Fig. 1b) in the gut. Our findings confirmed that an elevated fraction of IE-CTLs is present in CeD with a concomitant reduction in natural IELs. The latter population is only partially replenished in GFD patients, suggesting a slow recovery of natural IELs even after histological normalization of the hSI. We speculate that natural IELs might provide protection of the epithelial barrier at steady-state and that the relative proportional reduction of this subset may contribute to CeD pathology.

Several studies have proposed that viral infection or dysbiosis could be involved in CeD pathology[72,73]. In our study, we found that the

CeD-associated cells both in LP (CD4-C1 and CD8-C0) and in epithelium (IELs-C1) are associated with several viral/bacterial-related pathways in which the 'NOD-like receptor signaling pathway[74–76], including the common genes *IFI16* and *GBP2/4/5*, is unanimously upregulated in all the CeD-associated LP T and IEL cells (Supplementary Data 1). This could indicate that in addition to activated gluten-specific T cells, a large population of both CD4 and CD8 T cells in LP as well as the IELs, are triggered by viral/bacterial stimuli as a potential contributing mechanism in the course of the disease.

In conclusion, our atlas provides an important framework in expanding our knowledge about this complex immune-mediated disease and we provide a more comprehensive overview of the differences as a resource that could collectively hint to further follow up studies to find a best approach to manage or hopefully treat the disease.

## Methods

### Tissue samples

Duodenal biopsies were obtained during routine endoscopy at Akershus University Hospital and Oslo University Hospital from patients referred due to suspicion of CeD. Diagnosis of CeD followed standard procedure (Ludvigsson et al., 2014). 8 (all females, median age 29.7) patients with confirmed untreated CeD (Marsh score 3A-3C) and 7 (6 females, median age 35.8) confirmed non-CeD controls and 5 patients on gluten-free diet have been enrolled in the study. See Supplementary Table 1 for the list of patients and their clinical information. Biopsies have been placed immediately in ice-cold buffered saline, transported on ice, and processed within 4 h. The study was approved by the Norwegian Regional Committee for Medical Research ethics (REK 20521/6544), and all patients gave written informed consent and did not receive compensation.

### Preparation of single cells suspension

To separate the epithelium and IELs, biopsies were shaken twice in 6.5 ml of PBS with 2 mM EDTA (Sigma-Aldrich), 1% FCS (Sigma-Aldrich) and 1 μM flavopirydol (Sigma-Aldrich) for 10 min at 37 °C. Supernatants containing the epithelial fractions were combined, washed, passed through 100 μm cell strainers (Miltenyi Biotec), washed again and kept on ice until staining. Epithelium-free mucosa was minced and incubated with stirring in 2.5 ml of RPMI1640 (Lonza) containing 10% FCS, 1% Pen/Strep (Lonza), 1 μM flavopiridol[77], 0.25 mg/ml Liberase TL (Roche) and 20 U/ml DNase I (Sigma) for 40 min at 37 °C. Halfway through the incubation the samples were triturated using 1 ml pipette to facilitate complete digestion. Digested cell suspension was passed through 100 μm cell strainer and washed twice.

### FACS sorting

Cells were stained with a combination of fluorescent antibodies in FACS buffer for 20 min on ice, washed twice, and filtered through 100 μm nylon filter mesh before sorting. Both LP and epithelial fractions were stained with: FcR Blocking Reagent (Miltenyi Biotec), CD45-APC-H7 (2D1, BD Biosciences), CD3-APC (OKT3, Biolegend), CD19-BV421 (HIB19, Biolegend), HLA-DR-PerCP-Cy5.5 (L234, Biolegend), CD14-PE-Cy7 (HCD14, Biolegend), CD11c-PE (S-HCL-3, BD Biosciences). EpCAM-FITC (Ber-EP4, Dako) was added to the epithelial fraction to label epithelial cells. In some samples CD27-BV605 (O323, Biolegend) and CD103-BV605 (Ber-ACT8, Biolegend) were added to the LP and epithelium fraction, respectively. Dead cells were stained and excluded using To-Pro-1 (Molecular Probes) (see Supplementary Table 2 for more details on the catalogue numbers and the dilutions of the antibodies used in this study). Doublets were identified and excluded based on scatter parameters in forward scatter area versus height (FSC-A/FSC-H) and side scatter area versus width (SSC-A/SSC-W) plots. Cells of interest (single live CD45+, or

single live CD45+(CD27-)CD3-CD19- to enrich for myeloid cells) were index sorted (48 epithelial cells per patient and the rest LP cells) into Bio-Rad Hard Shell 384 well microplates (Bio-Rad) containing well specific primers (100 nl, 0.75pmol/ul) and 5 ul mineral oil (Sigma-Aldrich) on top using FACS Aria IIu or FACS Aria III with FACS Diva software version 8 (BD Biosciences) (see Supplementary Fig. 8 for the flow cytometry gating strategy). After sorting plates were sealed, labelled, span down for 10 min at 2000g, snap frozen on dry ice and kept at −80 °C until used. In addition, a minimum of $10^5$ total events from each sample were recorded and exported with index sort files for analysis in FlowJo v 10.6 software.

### scRNA-seq modified SORT-seq

In order to obtain the transcriptome of immune cells of hSI in CeD and normal controls, we used the SORTseq protocol (Muraro et al. 2016) which itself is a derivates of original CEL-Seq2 protocol (Hashimshony, T. et al 2016) and applied some modifications to the original protocols. In brief, single immune cells were first indexed with canonical markers of interest using the previously described antibodies and then sorted in 384-well PCR plates (BioRad). The plates were already primed with unique primers per each well in which each primer has a unique cell barcode with UMI incorporated to tag each RNA molecule in the cell. Sorted plates then were kept frozen at −80 °C before proceeding to cDNA synthesis. The frozen plates were shortly centrifuged at 250 x g for 1 min at 4 °C and incubated in the PCR machine at 60 °C for 2 min to break the cells and release the RNA.

We used micro-dispenser machine, Nanodrop II (BioNex) to dispend all the reagents in the next steps of library preparation. After cell breakage, 100 nl lysis buffer contains RNase inhibitor (RNasin PLUS promega), Triton X100, ERCC spike in control RNAs 1:50,000 diluted (Ambion) were added to each well, short spinned at 250 x *g* for 1 min and incubated in the PCR machine at 65 °C for 2 min and then immediately put on ice for 5 min.

Then first 150 nl cDNA reaction mix contains first strand cDNA synthesis buffer (Invitrogen), SuperscriptII reverse transcriptase (Invitrogen), 0.1 M DTT and RNasein PLUS (promega) was added to each well of the plate and the plates were spinned at 250 x g at 4 °C for 1 min and placed in PCR machine at 42 °C for 1 h followed by 10 min enzyme deactivation at 70 °C and immediately placed on ice.

Then, the 500 nl second strand DNA reaction mix contains second strand cDNA synthesis buffer (Invitrogen), 10 mM dNTP, E. Coli DNA ligase enzyme (NEB), E.coli DNA polymerase enzyme (NEB) and RNaseH enzyme (com) was added to each well of the plate and then plates were spinned for 1 min at 250 x g and placed in the PCR machines at 16 °C for 2 h followed by enzyme deactivation at 70 °C for 10 min and then placed on ice.

To remove the excess remainder of the primers in the reaction mix in each well, we used exonuclease-I enzyme (NEB) and incubated the plates at 37 °C for 20 min followed by enzyme deactivation at 85 °C for 15 min and immediately placed the plates on ice.

To reduce the DNA loss during the subsequent steps of library preparation, we linearly amplified the cDNA content of each well using unidirectional linear-PCR using Pfu DNA polymerase enzyme (Promega), forward primer (5′-GCC GGT AAT ACG ACT CAC TAT AGG GGT TCA G-3′) against the common sequence of the T7-P5 part of the original oligo-dT primers and ran the linear-PCR for 15 cycles with 2 min preheating at 95 °C, then sequential cycles of 30 s at 94 °C, 30 s at 50°C and 3 min at 72 °C and ended by incubation at 72 °C for 5 min and then placed plates on ice.

After linear amplification of the DNA content in each well, the reaction solution of all wells per each plate were pooled together (∼600 ul) and the DNA was cleaned up using AMPure XP beads by mixing 100 ul XP beads with 600 ul clean up buffer contains PEG and tween and adding it to the DNA pool. The DNA-bond beads then

washed two times with 80% ethanol and the DNA was eluted in 22 ul EB buffer.

Then the amplified single strand DNA fragments of each pool was second stranded using second strand DNA synthesis buffer (NEB), first strand cDNA synthesis buffer(NEB), 0.1 M DTT, random hexamer primer mix (Thermo), dNTP mix, E.coli DNA polymerase (NEB) and E.coli DNA ligase (NEB) in 50 ul reaction volume and incubated for 2 h at 16 °C. Then the DNA was cleaned up using 2x XP-bead:DNA ratio and eluted in 7 ul water.

Then the DNA was amplified and transformed to RNA using in vitro transcription overnight according to manufacturer's protocol (Ambion Mega-Script IVT kit) at 37 °C. The amplified RNA (aRNA) then was treated with ExoI-rSAP enzymes for 20 min at 37 °C and fragmented using Manganese-based fragmentation buffer for 1.5 min at 94 °C followed by incubation with 0.5 M EDTA buffer to stop the fragmentation process. The fragmented aRNA was subsequently cleaned up using 2x XP-bead:aRNA ratio and eluted in 12 ul water.

We used 5 ul of the aRNA and proceeded to next step of library preparation. The cleaned aRNA was converted to cDNA using 1 ul of random octamer primers that contain part of T7 sequence of Illumine library oligos and the reaction mix contains first strand cDNA buffer (NEB), 0.1 M DTT, 10 mM dNTPs, RNasein PLUS (Promega) and SuperscriptII reverse transcriptase enzyme (Invitrogen) was incubated at 25 °C for 10 min followed by incubation at 42 °C for 1 h and 70 °C for 10 min and then placed on ice. Then the final library was amplified and enriched by 7 cycles of PCR using KAPA hyper prep kit and adding forward and reverse primers against Illumine T5 and T7 sequences in which the reverse primer contained unique sequence and added specifically per each sample/library. The library then was cleaned with 1:1 XP-bead:DNA ratio and eluted in 20 ul elution buffer. The quantity of the library was measured using KAPA library quantification kit (Roche) and the quality was checked by using the bioanalyzer machine (Agilent).

### scRNA-seq data analysis

Each library obtained from one plate was sequenced for on average 40 million reads using NextSeq 500 sequencer. The raw FASTQ files (Read2 contains 8nt cell barcode $^+$ 8nt UMI; Read1 contains mRNA sequence) were used as input for the CEL-Seq2 pipeline (Hashimshony, T. et al. 2016) however, incorporating STAR aligner (https://github.com/alexdobin/STAR) to map the sequenced reads against the human genome HG38 using the default parameters. The uniquely mapped reads then were demultiplexed per each single cell per library using the known 384 cell barcodes. The demultiplexed, uniquely mapped reads per each single cell then were counted against the reference genome and PCR deduplicated using HTSeq-count part of the CEL-Seq2 pipeline and the table of read counts for all single cells was built using R environment.

To remove the low-quality cells from the dataset, we applied several criteria. Cells with MAD value of ERCC, mitochondrial counts and library size greater than 3 were removed using SCRAN package (v 1.4.5). Next, cells with total detected genes less than 300 and more than 4000 were assigned as outliers as they fell in the extreme tails of the normal distribution of gene complexity across the single cells (Supplementary Fig. 1c). The latter cells ($n = 91$) had gene complexities above the ranges of detected genes per each immune cell type in our dataset (Supplementary Fig. 1d) most likely as a result of cell aggregates during the cell sorting; hence the outlier cells were excluded from subsequent analysis.

We also removed the drop-out genes from the dataset; for this we removed the low-abundance genes with either zero expressions for all the cells or the genes with mean expression value below the threshold on the distribution model[78,79] (Supplementary Fig. 1g).

We included no-cell wells during the sorting of the cells in the plates and used these wells as background controls (empty-well controls) for removing the genes with high background noise to signal expression. To this end, we first assigned the common ERCCs that were detected between single cells and empty-well controls (cERCC) in our dataset. Then we calculated the values of cell expression and empty-well control expression per each gene divided by the cERCCs (cell/cERCC and Empty-well/cERCC); to make a fair comparison, we randomly subsampled the cells to the same number of the empty-well controls; performed 1000 times permutations and took the average. We then build a matrix of the means of the calculated values of all the cERCC and measured the ratio of empty-control (noise) divided by cell expression per each gene (noise/signal). We plotted the distribution of these ratios and set the threshold where we had the sharp drop in the distribution model and the genes with high background noise/signal ratios above the threshold were removed from the dataset (Supplementary Fig. 1h).

The dataset of filtered high-quality cells (~6250) then was normalized using size-factor calculations within SCRAN package (v 1.4.5) incorporating the ERCC spike-ins and used for further analysis. Beside subtle variations between donors, we did not observe major batch effects that would compromise the analysis and the identified clusters/cell types, and the resulted clusters were formed by contribution from several donors (Supplementary Fig. 1i). Hence, we did not perform any batch correction algorithm on the data.

Subsequently, we used Seurat v.3 pipeline (Butler et al. 2018) to identify the possible clusters of cells and the main cell populations using the Wilcoxon Ranked-sum test with the default parameters. Then using Seurat pipeline, we identified marker genes per each cluster of cells and performed the heatmap analysis accordingly in R environment using pheatmap package. For gene ontology analysis (GO), including KEGG pathway enrichment analysis, we used the identified gene signatures per clusters and fed the signatures into ClusterProfiler[80] pipeline in R environment with default parameters to perform the enrichment analysis. GO terms with Benjamini-Hochberg p-values of less than $10^{-3}$ were selected as significant terms.

The UMAP analysis was subsequently performed using the related integrated function within the Seurat package with the default parameters.

To assess the TCR expression pattern among the cells, we used the mRNA expression levels of the TCR constant region genes, *TRGC1*, *TRGC2*, *TRDC*, *TRAC*, *TRBC1* and *TRBC2*, and looked at the expression patterns between the cells given the fact that the TCRg/d cells have higher expressions of *TRGCs*/*TRDC* and TCRa/b cells express higher levels of *TRAC*/*TRBCs* genes.

To obtain the cluster markers for the related figures (Figs. 5 and 6), we applied a linear modeling analysis incorporating Bayesian statistics using Limma package (Ritchie ME et al. 2015) with the default parameters. For GWAS-based analysis, we used the publicly available catalog of CeD-associated SNPs from the NHGRI-EBI Catalog of human genome-wide association studies at https://www.ebi.ac.uk/gwas/ and assessed the mRNA expression of the genes which had at least one CeD-associated SNP, in our dataset.

To estimate the cell proportional changes between different clusters and cell types in our dataset, we did a permutation of 10 times random subsampling (50%) of the cells per each cluster and measured the percentage of the cells across all the related clusters per each condition (Ctrl, CeD and GFD) and plotted the results in boxplots. The p-value of the differences between the conditions was calculated using student t-test based on the result of the 10 random subsampling of cells per each cluster per condition.

For deconvolution analysis of the identified single-cell signatures on bulk RNAseq dataset, we used recent publicly available datasets from human healthy and CeD small intestinal biopsies with GEO

accession number GSE146190 and used the SCDC pipeline[81] with default parameters to perform the analysis.

## Statistics & Reproducibility

The statistical analysis of the flowcytometry data in this study was carried out using Prism 6.0 (GraphPad Software). The single cell RNA sequencing data generated in this study was analyzed for statistical significance of the differential gene expressions using the non-parameteric Wilcoxon rank sum test and/or empirical Bayes statistics accordingly. The significance of the differences in the proportion of the cell types between the conditions was determined by using student t-test (two-sided p value) statistics. For gene ontology analysis, the Benjamini-Hochberg FDR ($< = 10^{-3}$) was used to define the significant enrichment. No statistical method was used to predetermine sample size. Biological replicas have been used in this study per each condition (Ctrl, CeD and GFD) and the outcome was reproducible between the subjects within the conditions. Applying the quality control criteria mentioned in the previous sections, the sequencing data from low quality cells were excluded from this study. The experiments were not randomized; The Investigators were not blinded to allocation during cell isolation experiments and outcome assessment.

## Reporting summary

Further information on research design is available in the Nature Research Reporting Summary linked to this article.

## Data availability

The sequencing data generated in this study has been deposited at the European Genome-phenome Archive (EGA), which is hosted by the EBI and the CRG, under accession number EGAS00001003751. The raw sequencing data are available under controlled access (privacy regarding patient materials) by requesting permission from the corresponding Data Access Committee (DAC) [https://ega-archive.org/dacs/EGAC00001000557]. The bulk RNA sequencing data used in this study for comparison, is publicly available at the GEO database under accession number GSE146190. Source data are provided with this paper.

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

## Acknowledgements

We would like to thank Dr. Joost Martens at the department of Molecular Biology at Radboud University for his invaluable input and the fruitful discussions. NA, EJM and HGS are funded by the European Union grant ERC-2013-ADG-339431 'SysStemCell' and PB and HGS are funded by ALW 'Programming human macrophages' grant under dossier number ALWOP.2015.101. AB and EB are funded by grants from Southern and Eastern Norway Regional Health Authority (2013018/2013 and 2018044/2018).

## Author contributions

N.A., E.B., H.G.S and F.L.J. conceived the study and wrote the manuscript. N.A., A.B. and E.B. performed the experiments and analyzed the data. P.B. and E.J.M. contributed to experimental design. KEL and JJ coordinated the patient material collection. H.G.S. and F.L.J. supervised the study and H.G.S. is the corresponding author of this study.

## Competing interests

The authors declare no competing interests.
