## [Peer Review File · Nature Communications]

Single cell transcriptomic analysis of the immune cell compartment in the human small intestine and in Celiac diseaseREVIEWER COMMENTS

Reviewer #1 (Remarks to the Author):

Summary: The authors have conducted a very thorough characterization of the transcriptome of the small intestine of celiac disease patients, resulting in a large database (atlas) of information detailing prominent changes in the transcriptome. Unfortunately, there were no outstanding novel (noteworthy) results regarding pathogenesis of Celiac Disease from such a herculean effort. Most of the transcriptional changes observed by the authors were also pointed out by the authors to have been identified in previous studies, which is good in that this proves that their approach of using scRNA seq worked; yet, there are no results that truly stand out as novel with regard to Celiac Disease pathogenesis. Most of the manuscript is listing changes in obscure genes without giving context as to how these obscure genes could play a role in the development of Celiac disease. There are some hints to novelty though, such as the observed changes in the activation state of the CD8+ cells in the lamina propria of Celiac Disease, but the authors did not pursue this any further than the transcriptional level. If the authors were to choose one of these potentially novel pathways (maybe LP CD8+ cells?) identified in their study and do confirmational experiments at a protein level or higher, that would greatly increase the novelty and reduce the confusing aspects of the data presentation of the manuscript.

Major Criticisms:

- 1) In the Results section, lines 92-96, there needs to be more detail in describing how the authors distinguished between intraepithelial lymphocytes and lamina propria lymphocytes, because no mention is made here that this was done by a physical/mechanical means (EDTA), as opposed to FACS sorting.
- 2) Why wasn't an antibody against the gamma delta TCR used in the FACS sorting? This is an important IEL population and a number of reports have shown significant changes within the gd T cells of untreated Celiac patients (Eggesbo et al 2019 Mucosal Immunology). However, lines 298 and 300 say that $\gamma\delta$ TCR cells were evaluated? Please clarify in both the methods section and lines 298-300 as to how exactly $\gamma\delta$ TCR cells were evaluated.
- 3) In line 304, why wasn't IL-2 observed to be upregulated in untreated Celiac patients? There are many studies that show that gluten challenge will increase the levels of IL-2 in the blood of Celiac patients.
- 4) In the paragraph between lines 302 and 311, the text states that CeD associated with the cluster C0 (Fig. IL15 is known to be increased in a subpopulation of Celiac disease patients as compared to controls, and is found to render CD4+ T cells non-responsive in Celiac diseases; however, IL-15RA was not found to be increased in the C0 (CeD) population of CD4+ T cells. Rather, it was increased in the C2 population. Why?
- 5) Because none of the clustering events observed with CD4+ T cells (Fig. 4—C0 and C1) and CD8+ T cells (Fig.5--C0, C1, C2, C3) truly associated exclusively with Celiac disease (C0 vs C1 vs C2 vs C3), it is incredibly difficult to derive conclusions that provide novel insight into the pathogenesis of Celiac disease.
- 6) Similarly, in Fig. 4D, Measles is the KEGG pathway most associated with C1 of CD4+ T cells, yet the authors do not mention this in the results or discuss this in the discussion.
- 7) On line 340..."However, we do observed a higher presence of TCR g/d expressing cells within the CeD-dominated C1 which is in line with the previous findings that the TCR g/d IELs are increased in CeD (Fig. S6B)." , Fig S6B is stated as containing information on TCR g/d IELs. I could not find this in Fig. S6B.

Minor Criticisms:

- 1) The term leukocytes is in the title of Figure 6. It should be lymphocytes.

Reviewer #3 (Remarks to the Author):

This study investigated the CD45+ immune cells in normal subjects and patients with celiac disease before and after gluten-free diet using single cell transcriptomics. They identified a group of natural intraepithelial leukocyte cells that is depleted and observed elevated numbers of activated intraepithelial T cells in the celiac disease patients. The study provided a valuable resource for studying celiac disease at single cell resolution which could be really appreciated by the community. Overall, I think the study is interesting and brings new insight, and the manuscript is well written. However, I do have a few suggestions the author may want to consider improving their manuscript as detailed below.

Major:

1. The authors claimed in their introduction that 'the detailed mechanistic pathways of the disease are not fully clear', but the authors did not dig into the mechanism much themselves either rather than reporting a populational change of immune cells. I would suggest they investigate more into the potential mechanism of the disease based on their analysis result or make hypothesis based on in house observation and existing literature knowledge.
2. The author should validate the changes of cell population they reported based on the existing literature. This could be potentially done by finding the IEL specific marker and investigating the marker expression in bulk RNA seq data in celiac disease patient compared to healthy subjects. This could also be done by using deconvolution method on existing bulk RNA seq data.
3. Method about the data analysis is not clear, especially in terms of batch correction. In the UMAP plot, all the samples were integrated, which means that batch correction was performed. The method of correction could highly affect the result of clustering. For example, if the batch effect among conditions were corrected, the biologically meaningful difference could be removed as well.
4. Overall, the manuscript lacks a strong and clear biological message to deliver and feels very much like data presenting. The authors basically relied on their conclusions on clustering and marker genes. The author may want to investigate more into their data to obtain some potential mechanistic insight or functional relevance.

Minor:

1. The legend title of gene expression heatmaps are misleading, where it says "Mean mRNA expression" but contains minus values.
2. The abbreviations and expansions of some words are mixed. For example, intraepithelial lymphocytes and its abbreviation IEL, lamina propria and its abbreviation LP.
3. It is not clear to me why the author would like to remove cells which expressed more than 4000 genes in their analysis.
4. The authors claimed 'We also removed the drop-out genes from the dataset and also the genes with high expression ratio of background to signal using empty wells' in their supplementary methods, but no detail was provided.
5. It is not clear how the authors performed the KEGG pathway enrichment analysis.
6. The discussion needs to be enhanced.

REVIEWER COMMENTS

Reviewer #1:

We would like to thank the reviewer #1 for the insightful and invaluable review. We agree with the reviewer that our study provides an extensive resource atlas that adds more dimensions to the current knowledge on celiac disease. Assessment of immune cells at such a high-resolution shows the transcriptome programs and the related celiac-associated changes/subtypes in the disease. Some of the findings were in line with previous studies on bulk samples supporting our findings at single-cell level and we provide a more comprehensive survey on the topic. Hence, we believe that our results provide a valuable resource and basis for conceiving follow up studies on mechanistic characterization and functional assessments of the findings. Here, we would like to address the reviewer's issues and suggestions to the best of our ability as follows:

Major Criticisms:

1) In the Results section, **lines 92-96**, there **needs to be more detail** in describing how the authors distinguished between intraepithelial lymphocytes and lamina propria lymphocytes, because no mention is made here that this was done by a physical/mechanical means (EDTA), as opposed to FACS sorting.

We have provided the details of the intraepithelial and LP cell separation in the method section under "**Preparation of single cells suspension**". Now we also added the related text in the result section (**line 87**) to eliminate the confusion.

2) Why wasn't an **antibody against the gamma delta TCR** used in the FACS sorting? This is an important IEL population and a number of reports have shown significant changes within the gd T cells of untreated Celiac patients (Eggesbo et al 2019 Mucosal Immunology). However, lines 298 and 300 say that $\square\square\square\square$ TCR cells were evaluated? Please **clarify in both the methods section and lines 298-300 as to how exactly $\square\square$ TCR cells were evaluated.**

In the design of our study, we took the index-sorting approach by which we used a panel of antibodies against the canonical immune markers to acquire the immune-cell information during the cell sorting and we used the CD45 channel for sorting the cells randomly in the plates in an unbiased manner. As part of the experimental design, we used transcriptional inhibitor flavopiridol, which decreases the transcriptional initiation and elongation^{1,2}, to avoid transcriptome alteration during the sample processing. In this regard, for T-cells we only used anti-CD3 antibody and as precaution, did not use antibodies against TCR g/d/a/b to mitigate any unwanted conformational changes and possible induction of the TCR-signaling³⁻⁵ and subsequently technical alteration of the original transcriptome of the cells. Given that we used empirical whole transcriptome analysis in this study, we envisioned to obtain a multidimensional data/power to get information about the genome-wide transcriptome of the cells including the TCRg/d cells among the others. In our study, we assessed the mRNA expression of TCR constant region genes and we found that in most cases, the TCRg/d and the TCRA/b expressing cells intermingle forming one distinct cluster showing very similar transcriptome profiles. The g/d CD8 T cells in LP have received little attention and, in our

study, we report that the percentage of TCRg/d expressing cells are elevated in LP CD8 T_{RM} cells in cluster C3 (Supplementary Fig. 5b). Moreover, the CeD-dominated IEL CD8 T cells in cluster C1 (Supplementary Fig. 6b) also show increased percentage of TCRg/d expressing cells which is in line with previous findings that the g/d IELs are increased in celiac⁶. To elaborate on the TCR assessment, we now modified the text in the result section (**lines 247-249**) and explained the analysis in the method section (**lines 563-567**) accordingly.

3) In line 304, why wasn't **IL-2** observed to be upregulated in untreated Celiac patients? There are many studies that show that gluten challenge will increase the levels of IL-2 in the blood of Celiac patients.

In our study, we could not detect IL2 mRNA expression among the tissue-derived cells. Alternatively, the protein level might not correlate with mRNA expression of this particular gene⁷ or the mRNA expression might be very low that given the limitation of the technology, could not be detected among other genes⁸.

4) In the paragraph between lines 302 and 311, the text states that CeD associated with the cluster C0 (Fig. IL15 is known to be increased in a subpopulation of Celiac disease patients as compared to controls, and is found to render CD4+ T cells non-responsive in Celiac diseases; however, IL-15RA was not found to be increased in the Co (CeD) population of CD4+ T cells. Rather, it was increased in the C2 population. Why?

In CD8+ T-cells in LP, we found high expressions of **IL15 in CD8-C3** cells (T_{RM}). Cells in cluster **CD8-C2** which are the recent recruited cells from peripheral blood, **express relatively higher levels of IL15RA** which suggest that they are the receiving/responding cells of the secreted IL15 ligand (Fig.5d). This could explain the potential role of resident memory cells in celiac patients on gluten-free diet in attracting the peripheral blood lymphocytes in LP and enhancing the reactions upon gluten ingestion.

Regarding the CD4+ T-cells in LP, we did not detect significant expression differences at mRNA level between CeD-associated CD4-C1 and the control CD4-C0 cells for IL15 and IL15RA (see plot below).

5) Because none of the clustering events observed with CD4+ T cells (Fig. 4—C0 and C1) and CD8+ T cells (Fig.5--C0, C1, C2, C3) truly associated exclusively with Celiac disease (Co vs C1 vs C2 vs C3), it is incredibly difficult to derive conclusions that provide novel insight into the pathogenesis of Celiac disease.

In our study, we found that the difference between the conditions manifest itself in the proportion of the cells that together harbor a unique transcriptome program among the immune cell types. Previous studies on CeD, were focused on gluten-specific T cells, which accounts for a minor subset. However, in our study, we find that there is a dramatic shift in the transcriptome of a large population of both CD4⁺ and CD8⁺ T cells, which are not necessarily gluten-specific, suggesting that a large population of both cell types are involved in the disease. We agree with the reviewer that the gene signatures we detect might not be ‘black and white’ and exclusively specific to celiac disease given the evidence that in some extensive functional studies on gluten:HLA-DR interactions, it has been also shown that even the thought gluten-specific CD4⁺ T cells could be found in other autoimmune diseases unrelated to CeD⁹. This indeed add more complexity to the pathobiology of the disease. In our study, we provide a more comprehensive overview of the differences as a resource that could collectively hint to further follow up studies to find a best approach to manage or hopefully treat the disease.

6) Similarly, in Fig. 4D, **Measles** is the KEGG pathway most associated with C1 of CD4+ T cells, yet the authors do not mention this in the results or discuss this in the discussion.

In this part, we thought to mention in the text the general signaling pathways among the top enriched ones. But now we split the pathways into ‘general pathways’ (Fig.4d) and we provided the ‘infection’-related pathways in the supplementary (Supplementary Fig. 4c). Furthermore, we added the related text in the result section (**lines 220-223**) and in the discussion pointing to the evidence that the gut microbiome is changed in CeD which could explain the relevance/association of the findings. We further extended the text in the discussion section elaborating on the potential role of viral/bacterial triggers in the course of the disease given that all the LP-CD4-C1 and LP-CD8-C0 as well as IELs-C1 cells are associated with several viral/bacterial pathways in our gene ontology analysis (**line 388-396**).

7) On line 340..."However, we do observed a higher presence of TCR g/d expressing cells within the CeD-dominated C1 which is in line with the previous findings that the TCR g/d IELs are increased in CeD (Fig. S6B)." , Fig S6B is stated as containing information on TCR g/d IELs. **I could not find this in Fig. S6B.**

The mentioned Supplementary Fig. 6b plot, depicts the mRNA expression of TCR constant region genes among the IEL clusters C0-C3. Here we show that ~75% of C1 cells express TRGC2 and ~50% of C1 cells express TRDC in which the expression of both genes is higher in C1 cells as compared to C0, C2 and C3 cells. In the revision, we implemented a heatmap of the mean mRNA expression of the TCR constant region genes in the IEL clusters in the related plot (Fig.S6b) to further elaborate the mean differences between the clusters.

Minor Criticisms:

1) The term leukocytes is in the title of Figure 6. It should be **lymphocytes.**

We corrected this accordingly.

Reviewer #3:

We would like to thank the reviewer #3 for the constructive and valuable review and the comments. We appreciate the reviewer's point of view that our study provides a valuable resource for the scientific community. Here we would like to address the reviewer's comments point-by-point with the best of our ability as follows:

Major:

1. The authors claimed in their introduction that 'the detailed mechanistic pathways of the disease are not fully clear', but the authors did not dig into the mechanism much themselves either rather than reporting a populational change of immune cells. I would suggest they investigate more into the **potential mechanism of the disease** based on their analysis result or **make hypothesis** based on in house observation and existing literature knowledge.

We thank the reviewer for the comment. We believe that our study presents an extensive resource for the scientific community that would be very helpful in guiding the focus on the mechanistic characterizations and the related functional studies on basic science and clinical aspects on the topic. We extended the discussion section and discussed the potential role of viral/bacterial contribution to the mechanism of the disease given that all the CeD-associated T and IEL cells in our study (LP-CD4-C1, LP-CD8-C0 and IELs-C1) are associated with several viral/bacterial pathways and are unanimously upregulating their 'NOD-like receptor signaling pathway' which is involved in intracellular defense mechanism against infection pathogens (**line 388-396**).

2. The author should validate the changes of cell population they reported based on the existing literature. This could be potentially done by finding the IEL specific marker and **investigating the marker expression in bulk RNA seq data in celiac disease patient**

compared to healthy subjects. This could also be done by using **deconvolution method on existing bulk RNA seq data.**

We appreciate the reviewer's comment and we performed the additional analyses as requested. We took two recent studies^{10,11} on celiac as reference of bulk RNAseq and we investigated the expression of the DEGs obtained from the related bulk studies in our identified single-cell clusters (Supplementary Fig. 7a, b). We also performed SCDC^{12,13} deconvolution analysis using the single-cell transcriptome signatures from our study on the bulk RNAseq dataset comparing celiac vs control mucosal samples (Supplementary Fig. 7c). We further investigated the natural IEL signature that we identified in our study in the related bulk RNAseq dataset (Supplementary Fig. 7d). We observed the major differentially upregulated genes in bulk studies are associated with MF-DC and MC cells we identified. The deconvolution analysis was in line with our findings on MF and CD4 T cell proportional changes and we found that the natural IEL signature was upregulated in healthy control samples as compared to celiac tissues with some genes upregulated in celiac which could be associated with other stromal cells in the tissue (**lines 317-337**).

3. Method about the data analysis is not clear, especially in terms of **batch correction**. In the UMAP plot, **all the samples were integrated, which means that batch correction was performed**. The method of correction could highly affect the result of clustering. For example, **if the batch effect among conditions were corrected, the biologically meaningful difference could be removed as well.**

We thank the reviewer for bringing up this point. We are indeed aware of the fact that batch corrections could potentially remove the biological information as well. We can confirm that we have not performed any batch corrections in our study. In the design and the experimental workflow of our study, we made utmost efforts to mitigate any potential source of batch effects (i.e keeping consistency in the experimental protocols, reagents, sample processing etc) and reducing the number of PCR cycles. The used linear-amplifications further reduces the exponential exaggeration of unwanted technical differences. We have mentioned throughout the text that we do not observe any major batch effects in our data sets and we provide analysis plots in the related sections (e.g Supplementary Fig. 2a, Supplementary Fig. 4a, Supplementary Fig. 5a) showing that the identified clusters were formed by contribution from several donors indicating the integrity of the clusters and cell types regardless of the subtle donor-to-donor variations. The UMAP^{14,15} analysis is of course a dimensional reduction algorithm to present the multi-dimensional data in a 2-dimensional space without necessarily applying batch-correction algorithms in between. Hence, we are confident that all the differences and altered transcriptional programs that we report here in our study between the cell types, clusters and conditions are related to biological differences. We now extended the method section on data analysis accordingly to be clearer (**lines 549-552**).

4. Overall, the manuscript lacks a strong and clear biological message to deliver and feels very much like data presenting. The authors basically relied on their conclusions on clustering and marker genes. The author may want to investigate more into their data to obtain some potential **mechanistic insight or functional relevance.**

We appreciate the reviewer's comment. We understand that our study does not present mechanistical/functional characterization of the findings as we aimed to carry out an extensive investigation of the transcriptome of all the CD45+ immune cells in an unbiased manner in CeD with a high resolution at single-cell level and to build/present a comprehensive resource atlas to the scientific community and the experts in the field. We indeed investigated the relevance of our findings to potential biological processes and pathways (gene ontology analysis) based on the current knowledge in the field resulting in the identification of several pathways and functions associated with the different conditions (e.g. Fig.4d and Supplementary Data 1). This indeed asks for follow up directions to functional/mechanistic assessment of the findings.

Minor:

1. The legend title of gene expression heatmaps are misleading, where it says "Mean mRNA expression" but contains minus values.

The legends show the mean mRNA expressions which are z-score transformed. We added this additional information to the legends accordingly to elaborate on this.

2. The abbreviations and expansions of same words are mixed. For example, intraepithelial lymphocytes and its abbreviation IEL, lamina propria and its abbreviation LP.

We made sure that this is consistent throughout the text.

3. It is not clear to me why the author would like to remove cells which expressed more than 4000 genes in their analysis.

As part of the quality controls, we set the thresholds to remove cells with gene complexities that fall in the extreme tails of the distribution model of the gene complexity across the single cells (Supplementary Fig. 1c). The cells with gene complexity above 4000 were assigned as outliers as they fall in the extreme-right tail of the distribution model and their gene coverage was not in range of the detected gene coverages per each immune cell type in our dataset (Supplementary Fig. 1d). This few numbers of cells (n=91, ~1%) could be a result of cell aggregates during the cell sorting and hence were removed from subsequent analysis. We now added this explanation in the text in the method section (**lines 523-528**) to clarify on this.

4. The authors claimed 'We also removed the drop-out genes from the dataset and also the genes with high expression ratio of background to signal using empty wells' in their supplementary methods, but no detail was provided.

We appreciate the point brought to our attention by the reviewer and accordingly we expanded this part in the text in the method section under 'scRNA-seq data analysis' (**lines**

529-544) in more explicit details and we provided additional analysis plots in Supplementary Fig. 1g, h to elaborate on this part.

5. It is not clear how the authors performed the KEGG pathway enrichment analysis.

We used ClusterProfiler pipeline¹⁶ for gene ontology analysis, including the KEGG pathway enrichment analysis on the identified cluster signatures. We expanded this in the text in the method section accordingly (lines 557-560).

6. The discussion needs to be enhanced.

We enhanced/extended the discussion section accordingly.

REFERENCES

- 1 Lam, L. T. *et al.* Genomic-scale measurement of mRNA turnover and the mechanisms of action of the anti-cancer drug flavopiridol. *Genome Biol* **2**, RESEARCH0041, doi:10.1186/gb-2001-2-10-research0041 (2001).
- 2 Chen, R., Keating, M. J., Gandhi, V. & Plunkett, W. Transcription inhibition by flavopiridol: mechanism of chronic lymphocytic leukemia cell death. *Blood* **106**, 2513-2519, doi:10.1182/blood-2005-04-1678 (2005).
- 3 Alarcon, B., Gil, D., Delgado, P. & Schamel, W. W. Initiation of TCR signaling: regulation within CD3 dimers. *Immunol Rev* **191**, 38-46, doi:10.1034/j.1600-065x.2003.00017.x (2003).
- 4 Davis, S. J. & van der Merwe, P. A. The kinetic-segregation model: TCR triggering and beyond. *Nat Immunol* **7**, 803-809, doi:10.1038/ni1369 (2006).
- 5 Guy, C. S. *et al.* Distinct TCR signaling pathways drive proliferation and cytokine production in T cells. *Nat Immunol* **14**, 262-270, doi:10.1038/ni.2538 (2013).
- 6 Valle, J. *et al.* Flow cytometry of duodenal intraepithelial lymphocytes improves diagnosis of celiac disease in difficult cases. *United European Gastroenterol J* **5**, 819-826, doi:10.1177/2050640616682181 (2017).
- 7 Lamb, J. *et al.* The Connectivity Map: using gene-expression signatures to connect small molecules, genes, and disease. *Science* **313**, 1929-1935, doi:10.1126/science.1132939 (2006).
- 8 Hicks, S. C., Townes, F. W., Teng, M. & Irizarry, R. A. Missing data and technical variability in single-cell RNA-sequencing experiments. *Biostatistics* **19**, 562-578, doi:10.1093/biostatistics/kxx053 (2018).
- 9 Christophersen, A. *et al.* Distinct phenotype of CD4(+) T cells driving celiac disease identified in multiple autoimmune conditions. *Nat Med* **25**, 734-737, doi:10.1038/s41591-019-0403-9 (2019).
- 10 Dotsenko, V. *et al.* Genome-Wide Transcriptomic Analysis of Intestinal Mucosa in Celiac Disease Patients on a Gluten-Free Diet and Postgluten Challenge. *Cell Mol Gastroenterol Hepatol* **11**, 13-32, doi:10.1016/j.jcmgh.2020.07.010 (2021).
- 11 van der Graaf, A. *et al.* Systematic Prioritization of Candidate Genes in Disease Loci Identifies TRAFD1 as a Master Regulator of IFN γ Signaling in Celiac Disease. *Front Genet* **11**, 562434, doi:10.3389/fgene.2020.562434 (2020).
- 12 Dong, M. *et al.* SCDC: bulk gene expression deconvolution by multiple single-cell RNA sequencing references. *Brief Bioinform* **22**, 416-427, doi:10.1093/bib/bbz166 (2021).

- 13 Avila Cobos, F., Alquicira-Hernandez, J., Powell, J. E., Mestdagh, P. & De Preter, K. Benchmarking of cell type deconvolution pipelines for transcriptomics data. *Nat Commun* **11**, 5650, doi:10.1038/s41467-020-19015-1 (2020).
- 14 Becht, E. *et al.* Dimensionality reduction for visualizing single-cell data using UMAP. *Nat Biotechnol*, doi:10.1038/nbt.4314 (2018).
- 15 Dorrity, M. W., Saunders, L. M., Queitsch, C., Fields, S. & Trapnell, C. Dimensionality reduction by UMAP to visualize physical and genetic interactions. *Nat Commun* **11**, 1537, doi:10.1038/s41467-020-15351-4 (2020).
- 16 Yu, G., Wang, L. G., Han, Y. & He, Q. Y. clusterProfiler: an R package for comparing biological themes among gene clusters. *OMICS* **16**, 284-287, doi:10.1089/omi.2011.0118 (2012).

Reviewer #1 (Remarks to the Author):

The authors have adequately addressed all of my concerns.

Reviewer #3 (Remarks to the Author):

The authors successfully revised the paper based on the reviewer's comments and I suggest the publication of the manuscript in its current version.